# Synodality in the Reception of the Second Vatican Council and Development of the Pastoral Orientations of the Chilean Bishops' Conference, 1965–1985

**José Ignacio Fernández** 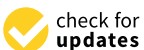

Facultad de Ciencias Religiosas y Filosóficas, Universidad Católica del Maule, Talca 3480112, Chile; jfernandez@ucm.cl

**Abstract:** After the Second Vatican Council, the Chilean bishops met in a plenary assembly during May 1968. As a result, the Episcopal Conference of Chile developed its first Pastoral Orientations (POs). Between 1968 and 1985, the Chilean Bishops produced eight different iterations of the POs. The ongoing development of the POs during this period reflected an emerging consensus across the various dioceses of the county. Five aspects of the POs in particular helped develop a synodal culture: (1) to reform the Church, (2) to establish the Church as an evangelizer and servant of humanity, (3) to opt for the base ecclesial communities as the local realization of the People of God, (4), to carry out the liturgical reform and diocesan synods, and (5) to develop intermediate forms of collegiality among the bishops.

**Keywords:** Second Vatican Council; synodality; Chile; People of God

## 1. Introduction

The reception process of the Second Vatican Council has the whole Church as its subject. It encompasses a wider reality than simply the actions of bishops. The Pastoral Orientations (POs), promulgated periodically between 1968 and 1985 by the Episcopal Conference of Chile, constitutes a first phase within this reception process in Chile[1]. We cannot consider that the reception process is limited to the POs. However, the POs offer insight into the wider ecclesial reception process, starting as they do from the episcopal conference, thus expressing the communion of Churches globally while also forming a local Church because they have been the way in which pastoral work has been organized (Cerda 2021, p. 13).

There is very little research on Chilean POs that we can point to until now. A work has recently been published by Alejandro Cerda that approaches POs from the topics they address (Cerda 2021). Particularly, the article by Sandra Arenas and Alejandro Zolezzi about the post-conciliar diocesan synods is an essential advance (Arenas and Zolezzi 2022). Other studies approach POs indirectly and do so from sociology or the history of the Chilean Church (Aguilar 2006; Huerta and Pacheco 1988) rather than from ecclesiology, as carried out in this study.

This research sought to recognize those ecclesiological aspects present in the POs text that promote a more synodal Church and that are the fruit of the conciliar reception in this period. In other words, are there permanent aspects in the POs of this period that help the realization of a more synodal Church? Are they a form of reception of Vatican II? Consequently, along with identifying those dimensions that are transversal to the set of POs and that allow a more synodal realization of the Church, their connection with the Second Vatican Council will be developed. To talk about synodality—or its adjective, synodal—we will take into account the well-known proposal of the International Theological Commission published in 2018:

In this ecclesiological context, synodality is the specific *modus vivendi et operandi* of the Church, the People of God, which reveals and gives substance to her being as communion when all her members journey together, gather in assembly and take an active part in her evangelising mission. (International Theological Comission 2018)

In fact, these POs show a range of participation in the drafting process. Over time, they increasingly included other ecclesial actors beyond the bishops themselves. The bishops also sought to collect and account for the path taken by the Church as a whole during the time preceding each consecutive promulgation of the POs.[2] The first POs were developed during the globally turbulent month of May 1968. At that time, the Chilean Bishops' Conference met in a Plenary Assembly to "clarify its ideas on the most important problems of the moment". The following year, a second set of POs were released. Between 1969 and 1972, at each Plenary Assembly, the Bishops' Conference promulgated a new document containing POs. Beginning in 1972, the POs were released with less frequency. By 1985, a total of 8 different sets of POs had been released by the Bishops' Conference[3].

The POs illustrate the process of reception of the Council. Five different aspects of this process suggest dimensions of a synodal realization of the Church. They may be seen as in continuity with the participation of the Chilean bishops in the Council and the post-conciliar experience. These five aspects are the reform of the Church, the Church as evangelizer and servant, the base ecclesial communities as the local realization of the People of God, the liturgical reform and diocesan synods, and intermediate forms of collegiality among the bishops.

## 2. Reform of the Church

The desire to reform the Church was not under discussion during Vatican II, as Pottmeyer points out, and instead, they discussed how to carry out this reform, changing the changeable and maintaining the immutable (Pottmeyer 2016). Consequently, the search for reform of the Church in the immediate post-Council should not surprise us. Thus, the aim of all of the policies and concrete decisions of this first phase of the POs was to give the Chilean Church a face in keeping with the image outlined by the Second Vatican Council[4]. In fact, the term "image" as applied to the Church is the axis around which the first POs were structured in 1968. They pointed out a path called "Towards the true image of the Church". This was centered on the potential of the base communities for their realization as the People of God: "The Father willed the salvation of mankind not in isolation, but by constituting a People. The primary objective of pastoral care is the formation of this People of God, the Church, the sacrament of salvation" (Episcopal Conference of Chile 1968; *Lumen Gentium* 1 and 9). In the same direction, the 1969 POs reinforced the pneumatological recovery of Vatican II,[5] pointing out that the Council presented a new image of the nature and mission of the Church as the sacrament of Christ, as an extension of the event of Pentecost. In response to this, and by a special movement of the Spirit, we are witnessing the rebirth of the Christian community (Episcopal Conference of Chile 1969).

In continuity with the Council's theological focus on the People of God, the synods held at the same time in Chilean dioceses offer dynamism to several POs. The bishops present the 1968 POs as being immersed in a circular relationship within the process of reform of the Church since the Council, pointing to the document itself as "a contribution of the bishops to the synods that are in progress throughout the country" and hoping that "once the diocesan Synods are over, we can perfect it and draw the definitive lines which will govern our Church in the years to come".

However, the explicit mention of diocesan synods diminishes toward the end of this stage: while appreciating that "the Pastoral Guidelines are the fruit of a long and methodical work, of a to-and-fro of the Episcopal Conference, the Pastoral Commission, and the National Organisms, at the grassroots and in the dioceses, and of many successive drafts" (Episcopal Conference of Chile 1981).

Consequently, a fruit of the process both recognized and further formed the People of God and their self-understanding. This is one of the peculiarities of the POs of this first phase: the Bishops' Conference of Chile understands itself in dialogue within the community of the baptized[6]. This awareness of the need for reform of the Church in a dialogue between pastors and the faithful is an aspect of ecclesial synodality. It is open to the future and the movement of the People of God in its historical journey toward the Kingdom of God.

### 3. Evangelizer and Servant of Humanity

The fact that the doctrine is oriented toward the vitality *ad intra* of the People of God, as is the case in the POs of this first phase, did not prevent the image of the Church sought from also carrying a vitality *ad extra*. In fact, the 1968 POs, in pointing out the path "towards the true image of the Church," indicates that the Church must be presented to all *as evangelizer and servant of humanity*. This binomial, in fact, is articulated in various ways as a permanent reference towards which the decisions and proposals made in the documents of this first phase are ordered[7]. More strictly, this binomial was used in the first four POs, as made explicit in those of 1971: "The center of all the activity of the members of the Church must be evangelization and the service of humanity".

In the field of evangelization, the POs of this phase are aware that Catholics considered themselves to be in a Christianity situation until recently: "We have taken the faith for granted" (Episcopal Conference of Chile 1968). Recognizing that the faith can no longer be taken for granted, the Chilean POs are advocating the reform of ecclesial structures, this time with an emphasis on sacramental pastoral care:

> it is absolutely essential to review the pastoral care of the preparation for the Sacraments. The greatest fruit that we wish to obtain from Baptism calls for a prior catechesis for parents which will enable them to fulfil their mission of initiating children into the Christian life. (Episcopal Conference of Chile 1968)

From the 1970 POs onwards, a tendency appeared which included the integral liberation of the human person as part of the evangelizing mission of the Church,[8] in line with the concluding document of Medellin.[9] In fact, it is evangelization that claims for itself the task of liberating the human being, understanding the proclamation of the Gospel as a liberating service, and thus integrating the binomial "evangelization" and "service to humanity". In fact, evangelization was understood as:

> Process or action of the Church to invite man to an experiential and explicit encounter with Christ the Saviour, present today in history, in individuals, and in the ecclesial community.

In relation to human promotion, it should not be confused with it. Evangelization is, however, committed to the action of integral liberation and human promotion in all its dimensions (Episcopal Conference of Chile 1970).

Integral liberation, as a form of service offered by the Gospel through the Church, along with the perspective of the *Medellin Document*, also finds an analogical continuity with the way Cardinal Silva Henríquez had defended religious freedom in the council meeting. Silva Henríquez had defended religious freedom in the conciliar hall[10]. In that debate, the Archbishop of Santiago had pointed to the Gospel as *oeconomia libertatis* (*Acta Synodalia* 1970–1999, IV/1, 227), from the development of the expression of Irenaeus of Lyons in *Adversus haereses* (Irenaeus of Lyon 1974, Lib. III, 10, 5).

The 1976 POs strengthened the understanding of evangelization as the priority task of the Church in the light of the publication of the exhortation *Evangelii Nuntiandi*. Both evangelization and the service of humanity were during this phase especially understood in an ecclesiological key. In other words, the Church continues being primarily subject and object of the POs. This primacy of the ecclesiological perspective for the evangelization and service can offers today insights for a synodal reform of the Church that avoids ecclesiocentrism.

#### 4. Base Ecclesial Communities

The Bishop Manuel Larraín had pointed out in the conciliar hall that the initiative to create the chapter on the People of God inside of the scheme *De Ecclesia* was the best way to express that "the pilgrim Church on earth is the mystery incarnated in the history of mankind"[11]. In continuity with this position, the Chilean Episcopal Conference explicitly developed its pastoral options in this first phase of the reception of the Council, starting from the category of the People of God. With this horizon as a point of reference for ecclesial life, the POs find in the base ecclesial communities a concrete form of participation in the Church as the goal of pastoral action. By way of example, the 1968 POs stated:

> The primary goal of pastoral ministry is the formation of the People of God, the Church, the sacrament of salvation.

The sacramentality of the Church is expressed in the universal Christian community and in the various local communities, presided over by a pastor in the name of the bishop, which in some way represents the universal visible Church.

The Bishops of the Episcopal Conference of Chile in 1968 wanted the primary pastoral work to be the formation of these Christian communities on a territorial or environmental basis. And this pastoral option became a constant in the POs of this first phase: "the promotion of these Christian Base Communities, together with the formation of Christian personalities, as our first pastoral priority" (Episcopal Conference of Chile 1969); "The Christian base communities have the mission of rebuilding the Church of today, starting from the bottom" (Episcopal Conference of Chile 1970); "That the Christian Base Communities be promoted as the basic nucleus to evangelize and serve humanity" (Episcopal Conference of Chile 1971); "In each ecclesiastical province, or as determined by the same province, a commission must be formed for the base communities or reflection groups" (Episcopal Conference of Chile 1973); the 1976 POs established five "priority areas of pastoral action," the first of these being the "base ecclesial communities" (Episcopal Conference of Chile 1975).

This communitarian form constitutes for the Chilean Church in this first post-conciliar phase the pastoral orientation par excellence for the integral reception of Vatican II. In fact, the effective emergence and dynamism of these communities were quickly understood as a sign of hope and of the action of the Holy Spirit in history. The 1969 POs express the following:

> By a special movement of the Spirit, we are witnessing the rebirth of the small Christian community. In many places and environments, communities of charity are springing up, the leaven of unity in their midst. They are born with the characteristics of the new image of the Church. They are like the fundamental cells based on which the Father wants to build and vitalize his people. (Episcopal Conference of Chile 1969)

In these words, it is already possible to recognize some elements that delineate the features of these communities: territorially or environmentally based groupings[12], which have Christ as their foundation and are linked in some way to the bishop, sacramentally reflecting the image of the Church in their place or environment, and which united like the basis cells of an organism, constitute the People of God through the work of the Holy Spirit.

The Chilean POs, as well as offering a theoretical framework as a horizon for the reception of Vatican II through the base ecclesial communities, can also be read as a real itinerary for the constitution, growth, and development of these groups. The POs of the first 20 years after the Council, seen in hindsight, unfolded a program of action that progressed from the theoretical framework through the recognition of its efficacy as a sign of the times to the adaptation and creation of ecclesial structures in accordance with it[13].

In the Chilean POs, the option for base communities does not appear to be detached from the personal dimension of Christian life. On the contrary, in a growing process, the personal formation of Christians is becoming more insistent, but it is presented within

the context of the base communities[14]. Moreover, from the 1973–1974 POs onwards, there is a tendency to repeat the binomial "the formation of individuals and of the community" (Episcopal Conference of Chile 1973) as inseparable pastoral actions, where one enhances the other[15]. However, in the 1982–1985 POs, the last of this phase, the precedence given to base ecclesial communities began to diminish, and this condition is now being shared within a set of six pastoral priorities[16]. Undoubtedly, the development of the base communities in Chile promoted the exercise of the common priesthood of the faithful, favoring the active participation of the baptized in the life of the Church through a sense of community belonging.

### 5. Liturgical Reform and Diocesan Synods

Bishop Larraín had advocated in the conciliar discussion that the common priesthood of the People of God should be developed theologically as a participation in the *triplex munus* of Christ (*Acta Synodalia*, II/3, 224), analogous to how the *De populo christiano* chapter of the *Chilean scheme* was structured.[17] However, the constitution *Lumen Gentium* omitted a discussion of the kingly aspect of priesthood from its chapter on the People of God, which focused only on the priestly and prophetic functions. Following this conciliar option, the Chilean POs of this first phase offer two ecclesial actions recognizable as ordered to the realization of the priestly and prophetic function of Christ: liturgical reform and the celebration of the diocesan synods.

The 1968 POs offered guidelines for the implementation of the liturgical reform in the particular case of the base ecclesial communities, thus interweaving different elements of the complex process of the reception of the Council: "The base Christian community needs a simple liturgy in which it can actively participate" (Episcopal Conference of Chile 1968).[18] The Chilean POs now offered some more concrete lines of action to facilitate and improve the participation of the faithful in the liturgy, such as "having, at certain times of the year, celebrations for youth groups, with expressions adapted to their character, familiar and intelligible, and with a clear historical meaning" (Episcopal Conference of Chile 1971). Perhaps one of the most significant ways toward the active and fruitful participation of the faithful in the liturgy, as an exercise of their common priesthood, is to be found in the renewal of sacramental catechesis at the family level. This is indeed progress and shows the Church in a state of mission rather than in Christendom.[19]

As for synods, they proliferated in the Chilean Churches during this first phase of reception of the Council: 13 of the 24 dioceses held synods between 1965 and 1985.[20] Although synods are not the subject matter of the POs, they are pointed out as a reference for their elaboration. In particular, the first POs of the Episcopal Conference of Chile not only recognize the fact of the various synods in the country but also place themselves in a relationship of mutual enrichment with them:

> The document that we offer today is a contribution of the Bishops to the Synods that are in the process of taking place throughout the country. [...] It will serve as a guideline until, after the Diocesan Synods, we can refine it and draw the definitive lines that will govern our Church in the years to come. (Episcopal Conference of Chile 1968)

Thus, the POs understood themselves at the beginning of this phase as a starting point toward the diocesan synods, to which they offer themselves as a contribution and, at the same time, as a point of arrival. It was a mutual reception between the POs of the Chilean Episcopal Conference and the synods of the particular Churches of the country. This proposal for a process of mutual reception gives rise to the idea of an episcopal authority that, collegially and locally, tries to accept in its actions the consensus reached in the synods of each diocese as a manifestation of the sense of the faith of the holy People of God, and anointed by the Spirit of truth, it also participates in the prophetic function of Christ (Episcopal Conference of Chile 1968). Consequently, what the 1968 POs say about the relationship of reciprocal reception between them and the diocesan synods manifests in

the need to involve the whole People of God in the journey by which they deepen the faith received and "apply it fully to life".[21]

Thus, the whole approach of this first phase of POs reveals the intention to prompt the active participation of the whole People of God in the exercise of the priestly function in the liturgy and its prophetic function through the diocesan synods.

## 6. Intermediate Forms of Exercising Collegiality

Already, the conciliar discussion had shown the positive assessment of the Chilean positions toward the advancement of the function of the Bishops' Conferences as an expression of the collegial request of the bishops for the Church. During the Council, Cardinal Silva Henríquez, while dealing with the *De oecumenismo* scheme, took into account the diversity of Protestantism in Latin America with regard to other regions. On this occasion, he considered it appropriate that it should be the episcopal conferences that should generate adapted norms that respond to the particularity of the situations (*Acta Synodalia* II/6, 72). This proposal was made in the context of the presentation of the so-called analogical exercise of ecumenism. Depending on the circumstances, the episcopal conferences were placed at an intermediate level of episcopal collegiality, which allowed for the adaptation of criteria to a particular reality. Likewise, when a footnote in the scheme *De Sacra Liturgia* threatened to restore to the Holy See the attributions on the particular application of principles in this field (*Acta Synodalia, Schemata constitutionum et decretum* I, 155), which the scheme gave to the episcopal conferences, the Archbishop of Santiago appealed for it to be withdrawn,[22] as was finally done.

Manuel Larraín, President of the Latin American Episcopal Council (CELAM), pointed out in an address at the third conciliar session that the development of collegiality in the episcopal conferences, in unity with the Supreme Pontiff, was a dynamism that expressed and strengthened the communion of the Churches, open to its exercise at the territorial levels (*Acta Synodalia* III/6, 555). Thus, articulating mission, collegiality, and communion, Larraín appealed to the intermediate exercise of collegiality in the regional episcopal conferences as a service to the ecclesial mission, using CELAM as an example.

The beginning and establishment of a new form of written expression by the Chilean Bishops' Conference in the immediate post-Council period, through the periodical promulgation of its POs, manifests the understanding that the bishops had of their common responsibility or co-responsibility in the Chilean Church. In fact, Chilean Bishops had already put into practice a few years before the Second Vatican Council, after the approval of its statutes in 1957, the common publication of texts. In these texts, the Conference expressed agreed upon positions on certain aspects of ecclesial reality. However, each of these documents responded to a different objective and lacked periodicity or common aims, unlike the POs.

The post-conciliar Chilean POs reflect the awareness matured at the Council that the "concern to proclaim the Gospel to all peoples belongs to all pastors since they all received together with the mandate of Christ which imposed on them a common duty" (Lumen Gentium 23c). Let us recall that this constitution also promoted the fruitfulness of the common solicitude for the Churches on the part of the bishops of the same region, recognizing the similarity that, in this respect, the conferences could have in comparison with the ancient groupings around a patriarchal Church (*Lumen Gentium* 23d). The Plenary Assembly of the bishops held in Chillán (1968), which gave rise to the first POs, reflected in them the atmosphere in which the meeting had taken place: "in the climate of the Second Vatican Council and of the Diocesan Synods and in the spirit of collegiality of the present time" (Episcopal Conference of Chile 1968).

That same year (1968), at the Medellin Conference, which brought together for the second time the Latin American episcopate, different lines of thought and action, which at the reception of Vatican II had begun to be aroused on this continent, converged. Except for the first POs of the Chilean Episcopate, which was promulgated a few months before the Medellin Conference,[23] the following Chilean POs of this phase found in the Medellin

conclusions an explicit reference point for priority options: the ecclesial base communities[24] and the beginning of processes of change in ecclesial structures.[25] This rich reception of Medellin within the Chilean POs in this phase puts two intermediate levels of episcopal collegiality—Latin American continental and Chilean national—into a relationship of enrichment, giving fruitfulness to the common concern for the ecclesial mission of the bishops of a given territory. Currently, theologians Rafael Luciani and Pedro Benítez see the collegiality of the Medellin Conference as a synodal event, as the emergence of an ecclesial way of proceeding (Luciani 2018, p. 512) and for the conception of the Church that is offered in the conclusive document (Benítez 2022, p. 509). In an analogous way, we can say this about the work carried out through the POs at that time by the Episcopal Conference of Chile.

In this spirit of common concern for the Churches, the POs of this phase, especially the first five (1968–1974), communicated common policies and decisions, as well as the means for their implementation in the Chilean Church. For example, in the 1968 POs, after receiving the approval of the Holy See for the implementation of the permanent diaconate in Chile, a plan of studies for the formation of future deacons was established.[26] The 1970 POs, after resolving that the "Episcopal Conference, through its Pastoral Commission, will pay special attention to the working class, rural and university environments" (Episcopal Conference of Chile 1970), make specific proposals separately for Catholic Action, the base ecclesial communities, and the parishes in Chile.[27] The 1971 POs, taking into account the national reality, list actions such as preparing "a family Catechism and writing it according to various cultural levels" and "reviewing the institutions of the Church, particularly the structures which have the greatest functional use" (Episcopal Conference of Chile 1970). All of these actions are common to the Churches of this country.

This plan of action by the Chilean bishops shows a dynamic wherein consensus is reached collegially for a common implementation of decisions regarding the mission in a territory determined by the Churches of Chile. This form of pastoral management avoids exaggerated centralism, with the consequent damage to the principle of subsidiarity, and disposes the bishops to a common concern for the good of the Church.

## 7. Conclusions

The five theological–pastoral aspects mentioned here begun to change at the end of the 1968–1985 phase and from then onwards in the Chilean POs. Broader doctrinal developments and a sociological turn began to appear. In fact, this made society, rather than the People of God, the object and the addressee of the POs. References to the general conferences of the Latin American episcopate, which expressed the common concern of the bishops at the local level, tended to diminish the same as those at diocesan synods, and those referring to the pontifical magisterium started to increase (Fernández 2023, pp. 390–405).

The aspects presented do not exhaust the conciliar reception of this first phase of the Chilean POs. In each of them, it is possible to recognize both a significant continuity with some theological lines of Vatican II and pastoral options, which tended to make the Chilean Church more synodal. In fact, the POs, rather than repeating the conciliar doctrine, are directed toward the generation of adequate structures in order to constitute an operative implementation of the pastoral purpose of the doctrine. In this way, the life of the Church itself is presented as a program of action (Some Bishops of the Episcopal Conference of Chile [1963] 2014, pp. 71–72), i.e., as an effective contribution to the *modus vivendi et operandi* of the Church.

The POs of this phase show that the category "People of God," with a high understanding of the role of the episcopate, is critical for the model of the Church advocated by the POs. The idea of a Church as the People of God is brought to fullness at a particular level through its bishop and is incorporated into the universal Church through his communion with his peers. At the local level, the common concern for the care of their Churches is exercised through the Bishops' Conference of Chile and CELAM. At the same time, the so-called

"base ecclesial communities" came to creatively realize the experience of constituting the People of God at the level of small territories within the parish structure, generating bonds that converge both in the community liturgy—promoting liturgical reform—and in participation in the evangelizing mission and the service of humanity. In these three dimensions, explicitly indicated by the POs of this phase, it is possible to witness the participation of the ecclesial community in the priestly, prophetic, and kingly functions of Christ, respectively; to which, at the level of the particular Church, the celebration of diocesan synods, also as an expression of the prophetic function of the People of God is added. This process is not a historical curiosity or a matter of nostalgia. It is nothing less than a way of proceeding for a more synodal Church in the ongoing reception of Vatican II.

**Funding:** This research received no external funding.

**Data Availability Statement:** Not applicable.

**Conflicts of Interest:** The author declares no conflict of interest.

## Notes

1. In fact, when we talk about reception in ecclesiological terms, a valuable reference is found in Congar's definition: "As 'reception' we mean here the process by which an ecclesial body in truth makes its own a determination that it has not given to itself [. . .] Reception is not the pure and simple realization of the relationship 'secundum sub et supra'; it includes a specific contribution of consent, possibly of judgment, where the life of a body which exercises original spiritual resources is expressed". (Congar 1972). This definition is also used to talk about the reception of Vatican II by Santiago Madrigal (Madrigal 2021).

2. For example, the 1968 POs say that "the document that we offer today is a contribution of the bishops to the synods that are in progress throughout the country". The 1973–1974 Pos state that they are "the fruit of the Pastoral Seminars" [carried out in the three preceding years] and of those Bishops' Assemblies. By 1982–1985, they noted that "these pastoral guidelines are the fruit of a long and methodical work, of a coming and going of the Episcopal Conference, the Pastoral Commission and the National Organisms, to the bases and to the dioceses".

3. I have used the division of POs into two phases and the aspects that characterize the first from a study already published (Fernández 2023, pp. 369–405). The division into these two phases arises from the change seen in the aspects of one and the other, so that they cannot be studied in the same phase. While in the previous study they sought to reflect the continuity/discontinuity with the Chilean participation in the Council, on this occasion, they try to show both their contribution to the realization of a synodal Church in the particular path of conciliar reception in Chile.

4. It should be recognized with Congar that the reform of the Church does not occur at the level of ideas but at the level of its historical forms (Congar 2014, pp. 163, 465).

5. The *Chilean Scheme*, presented as an alternative to *De Ecclesia* during the first conciliar session, similarly insisted on the trinitarian structure of the mystery of the Church, particularly the role of the Holy Spirit in its form, mission, charisms, and ministries in the hope of the eschatological realization of the Kingdom (Some Bishops of the Episcopal Conference of Chile [1963] 2014, pp. 82–83).

6. As several of their introductions of the POs make explicit: "The Chilean Episcopate, as a way of activating conciliar renewal in our country, have been providing pastoral guidance to the People of God since 1968" (Episcopal Conference of Chile 1973); "We have written them with pastoral agents in mind: community animators, catechist guides, youth advisors, ministers, laity, men and women religious, deacons and especially priests. They are the first recipients of this text, and also the first responsible for sharing these guidelines with their communities and, together with their respective bishops, seeing how to apply them in each of the country's dioceses" (Episcopal Conference of Chile 1981, N° 22).

7. In various ways, the POs articulate and develop the binomial "evangelizing" and "servant of humanity" applied to the Church. For example: ". . . Christians who serve the world and constitute a living nucleus of apostolic and charitable initiatives" (Episcopal Conference of Chile 1969); "The Chilean Church has taken in recent years a path of maturation in which two great lines are marked. The first line looks at its internal development, emphasizes catechesis and liturgy and is oriented towards the formation of Christian Base Communities. The second line looks at the evangelizing presence of the Church in the different human environments: worker, university, peasant, employee, professional, teacher, etc., and leads to proposing the Church-World commitment" (Episcopal Conference of Chile 1970).

8. "Christ's redemption wants to reach man in all his reality and the human community in all its expressions. By freeing men from sin, Christ wants to free them from all the slavery that comes from him" (Episcopal Conference of Chile 1969).

9. "Only in the light of Christ is the mystery of man made clear. In this light, all divine work, in the History of salvation, is an action of human promotion and liberation, whose only motive is love" (Second General Conference of the Latin American Episcopate 1968, I.II.4).

10. During the conciliar discussion, the Chilean positions were not homogeneous on the matter (See Fernández 2023, pp. 284–302).

[11]    "*Ecclesia peregrinans in Terris est mysterium incarnatum in hominibus historiae*" (*Acta Synodalia* II/3, 233).

[12]    Beyond the theoretical understanding studied here, the phenomenon of base Christian communities effectively developed and achieved not only ecclesial but also social significance (Bustamante 2009).

[13]    In the 1968 POs, successive stages are proposed: conversion, Christian initiation, prayer, sacraments; the 1969 POs confirm the birth of these communities as a motion of the Spirit and develops a description of their birth and growth; in the 1970 POs, there is an invitation to train promoters of base Christian communities (CEB) in each diocese and to hold diocesan or provincial meetings to review their progress; the 1971 POs point out the need to continue with the creation of these communities; the 1973–1974 POs determine the formation of a commission for the base communities in each ecclesiastical province; in the 1976 POs, people and families are called to join these communities, determining them as communities of faith, worship, love, and evangelization; the 1982–1985 POs establish "that in the Dioceses or areas where there is no Department or Commission in charge of the communities and their ministers (COMIN), a CEB training and promotion team be created as soon as possible" 1982–1985 POs, second part.

[14]    "The future of the base Christian community will depend mainly on deep and intensive pastoral care. This occurs in personal contact, spiritual direction, conferences, retreats on every occasion in which dialogue is conducive to successive conversions of the heart towards a greater perfection of love" (Episcopal Conference of Chile 1968); "It is important that it truly be a community of people, in which each one finds an enriching and generous path that allows them to grow in their love for the Lord and in service to others" (Episcopal Conference of Chile 1969).

[15]    "It is essential that Christians join and participate in the life of the Church" (Episcopal Conference of Chile 1975); as the last POs of this phase, summarizing the path followed, rightly recognize: "In the recent past we have invited people to form Christian communities, to form themselves as people and to concern themselves preferably with the poor" (Episcopal Conference of Chile 1981).

[16]    The youth, the family, the base ecclesial communities, education, the pastoral care of the multitudes, and the popular and marginalized sectors (Episcopal Conference of Chile 1981, second part).

[17]    After pointing out the common vocation of every baptized person to the Church and its mission, the *De populo christiano* chapter of the *Chilean Scheme* was structured as *populus sacerdotalis*, *populus apostolicus*, and *populum regalis* (Some Bishops of the Episcopal Conference of Chile [1963] 2014, pp. 122–24).

[18]    The 1975–1976 POs make this link again, pointing to the liturgy as one of the four necessary conditions for the base ecclesial community: "That they be a community of worship, in which the culmination of life is manifested in the celebration of the Eucharist" (Episcopal Conference of Chile 1975).

[19]    The need for a longer and deeper catechesis in preparation for the sacraments responds to the fact that the disposition to receive them can no longer be assumed or given by the environment, as in a state of Christendom, together with favoring the evangelization of the family environment, in the sense missionary.

[20]    The V synod of Ancud (1968); the I of Antofagasta (1968); the I of Araucanía (1968); the I, II y III of Chillán (1969, 1970 y 1971); the VI of Concepción (1968); the I of Linares (1967); the I of Osorno (1967); the II of Puerto Montt (1969); the I of Rancagua (1967); the VIII of Santiago (1967); the I of Talca (1969); the I of Temuco (1968); the I of Valdivia (1969). (See Gragnani 2015; Arenas and Zolezzi 2022).

[21]    "*in vita plenius applicat*" *Lumen Gentium* 12.

[22]    "*Est quia in instauratione s. Liturgiae non tantum S. Sedes sed etiam Conferentiae episcoporum partes habebunt, ut in ipso schemate proponitur*" *Acta Synodalia*, I/1, 609.

[23]    The first Chilean POs date from May 1968, while the Medellin Conference took place in September 1968.

[24]    The 1969 POs annex the document *Experiences in Chile on Christian base communities*, noting that it is carried out following Medellin's document, which recommends that theological, sociological, and historical studies be carried out on these communities.

[25]    "The Episcopal Assembly analyzed the real situation in terms of evangelization, catechesis and sacramentation, aware of the need to adapt the Church to the process of change that Medellin suggests. The Church must face this situation with suitable pastoral structures, renewing them to meet the demands of concrete situations, but with eyes fixed on the nature of the Church, the mystery of communion in Christ" (Episcopal Conference of Chile 1968).

[26]    It is considered that they will live from their work and that, "as a general rule, the Diaconate will only be given to married men, having the desire to leave the ordination of singles limited to very qualified cases" (Episcopal Conference of Chile 1968).

[27]    Proposals such as: "the Pastoral Commission of bishops dialogues with leaders and national advisers of the Catholic Action"; "train in the dioceses Christian case communities' promoters"; in anticipation of fewer priests "form teams of priests and nuns whose task is to educate, in a particular time, the laity who can continue to support the parish community. Centralize the administration of some sacraments. Centralize files". (Episcopal Conference of Chile 1970).

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
