# Peer review of "Synodality in the Reception of the Second Vatican Council and Development of the Pastoral Orientations of the Chilean Bishops’ Conference, 1965–1985"

_religions, doi:10.3390/rel14111374_

Round 1

Reviewer 1 Report

Comments and Suggestions for Authors

This is a very interesting and helpful piece. It is timely and has much to contribute to the ongoing ecclesiological conversation regarding synodality, its implementation, and its effects. It addresses a gap in knowledge. I enjoyed reading it.

The comments below are offered so that the overall argument might be strengthened and clarified.

I recommend a little more development in the introduction to clarify the argument from the beginning.

The very first paragraph introduces a lot of important ideas, many of which could be expanded. The overall argument of the article was not clear to me until p. 13. It would help to foreground key points such as the role of the people of God, and the orientation of the reception towards structure and action in the Introduction.

The theoretical framework can be strengthened in the introduction with more precise definitions. Provide a definition and more regional and ecclesiological context for the Pastoral Orientations (some of this is in the footnotes). Provide a definition of reception along with a little more theological context. Provide a definition of synodality and what you mean by synodal cultureperhaps with reference to the 2018 ITC document. With the definitions in place, the points can be linked to them, and the argument can be strengthened.

Overall, some references to secondary literature might be a helpful supplement.

In many cases the connection between the POs and the Second Vatican Council could be strengthened with more footnotes to the conciliar documents themselves.

A conclusion separate from the final point would be helpful. To some degree it is already there, but it will clarify to label it “Conclusion” so that the conclusions are distinct from the section about collegiality. 

Comments on the Quality of English Language

Moderate editing will be helpful. In some places the text relies heavily on quotations.

Author Response

Thank you for your valuable comments. Your report helps me to improve the article.

I have incorporated the following changes based on your review:

- I provide an idea of Pastoral Orientation in regional and ecclesiological context in the introduction (Cerda 2021).

- I provide a definition of reception in the first footnote (Congar 1972).

- It is referred the ITC’definition of synodality (2018).

- Contextualization with respect to theoretical background and other studies.

- Better explanation of the theological/ecclesiological argument in the introduction.

- Indication of the methodological aspects used.

- Engagement with more relevant literature.

- Extracting references from footnotes and putting them in the reference list.

- Distinguish the conclusions with a subtitle to offer greater coherence with the argument stated in the introduction.

All modifications were highlighted in blue (Please see the attachment).

Reviewer 2 Report

Comments and Suggestions for Authors

The paper brings contextual regional insights to the global contemporary Catholic and ecumenical discussion regarding synodality. This is the obvious strength of the paper. However, the method and the research plan should be explicated in the introduction to the theme. Also the conclusion part could be extended and reflected more. Now, the paper is not coherent and analytical enough. For example the significance of the Medellin conference could be explained more and to relevant research literature should be referred more. There are some spelling mistakes which should be corrected. I hope the work can improve these standard academical features, because the topic is interesting and relevant for the ongoing discussion and implementation.

Author Response

Thank you for your valuable contributions to improve the article.
I have accepted all of them. For this, I have incorporated the following changes:

- Contextualization with respect to theoretical background and other studies.

- Better explanation of the theological/ecclesiological argument in the introduction.

- Indication of the methodological aspects used.

- Improvement of the explanation of the significance of Medellin Conference.

- Engagement with relevant literature.

- Extracting references from footnotes and putting them in the reference list.

- Distinguish the conclusions with a subtitle to offer greater coherence with the argument stated in the introduction and with the development of the research. In turn, this allows us to better understand the contribution of this article to the existing literature.

All modifications were highlighted in blue (Please see the attachment). 

Reviewer 3 Report

Comments and Suggestions for Authors

I am attaching the article with a few formal comments.

Although the article is well-written and enjoyable, it is mostly a description of the Chilean Orientaciones pastorales between 1968-1985. I cannot grasp the main argument of this article: is it that the Chilean church has progressively and correctly received/applied Vatican II synodality/collegiality? There is no conclusion that helps to sum up all the information in the text.  There is no engagement with any kind of existing literature (if you look at the footnotes, there are no books or academic articles, just quotations from Orientaciones pastorales and Acta synodalia...). Neither I can grasp the methodology: is this a historical paper? A theological one? A sociological one?

I think that the author should strengthen the argumentative structure of his/her article, thinking of what new can this writing add to the already existing literature.

Suggested reading: Aguilar, A Social History of the Catholic Church in Chile

Author Response

Thank you for your valuable contributions to improve the article.
I have accepted all of them. For this, I have incorporated the following changes:

- Better explanation of the theological/ecclesiological argument in the introduction.

- Indication of the methodological aspects used.

- Indication of other studies in this regard.

- Engagement with existing literature.

- Extracting references from footnotes and putting them in the reference list.

- Distinguish the conclusions with a subtitle to offer greater coherence with the argument stated in the introduction and with the development of the research. In turn, this allows us to better understand the contribution of this article to the existing literature.

All modifications were highlighted in blue (Please see the attachment). 

Round 2

Reviewer 2 Report

Comments and Suggestions for Authors

Thank you for revising the paper. Now it is a good peace of research which will contribute to the discussion further regarding synodality. In the presentation of the keywords there is a spelling mistake: "sinodality"

Author Response

I have corrected the spelling of synodality in the presentation of the keywords.

Thank you for your contributions.

Reviewer 3 Report

Comments and Suggestions for Authors

The author improved his/her paper. He/She engaged with the existing literature, although superficially. The aim of the paper is now precise.

Author Response

The direct literature about pastoral orientations is not enough to go deeper into the dialogue with it, to which we add that the emphasis of the article is precisely on the work with the sources.

Thank you for your contributions.
